# Tb^3+^/Eu^3+^ Complex-Doped Rigid Nanoparticles in Transparent Nanofibrous Membranes Exhibit High Quantum Yield Fluorescence

**DOI:** 10.3390/nano10040694

**Published:** 2020-04-06

**Authors:** Peng Lu, Yanxin Wang, Linjun Huang, Sixian Lian, Yao Wang, Jianguo Tang, Laurence A. Belfiore, Matt J. Kipper

**Affiliations:** 1Institute of Hybrid Materials, National Center of International Joint Research for Hybrid Materials Technology, National Base of International Sci. & Tech. Cooperation on Hybrid Materials, College of Materials Science and Engineering, Qingdao University, Qingdao 266071, China; 18753360989@163.com (P.L.); newboy66@126.com (L.H.); 17854263619@163.com (S.L.); wangyaoqdu@126.com (Y.W.); 2Department of Chemical and Biological Engineering, Colorado State University, Fort Collins, CO 80523, USA; laurence.belfiore@colostate.edu (L.A.B.); matthew.kipper@colostate.edu (M.J.K.); 3School of Biomedical Engineering, Colorado State University, Fort Collins, CO 80523, USA; 4School of Advanced Materials Discovery, Colorado State University, Fort Collins, CO 80523, USA

**Keywords:** thin membrane, chemical synthesis, fluorescence, electrospinning, AFM

## Abstract

In this study, transparent membranes containing luminescent Tb^3+^ and Eu^3+^ complex-doped silica nanoparticles were prepared via electrospinning. We prepared the electrospun fibrous membranes containing Tb(acac)_3_phen- (acac = acetylacetone, phen = 1,10-phenanthroline) and/or Eu(tta)_3_phen- (tta = 2-thenoyltrifluoroacetone) doped silica (M-Si-Tb^3+^ and M-Si-Eu^3+^) and studied their photoluminescence properties. The fibrous membranes containing the rare earth complexes were prepared by electrospinning. The surface morphology and thermal properties of the fibrous membrane were studied by atomic force microscopy (AFM), thermogravimetric analysis (TGA) and differential scanning calorimetry (DSC), respectively. Fluorescence spectroscopy was used to characterize the fluorescence properties of the membranes. During the electrospinning process, the PVDF transitions from the α phase to the β phase, which exhibits a more rigid structure. The introduction of rigid materials, like PVDF and silica, can improve the fluorescence properties of the hybrid materials by reducing the rate of nonradiative decay. So the emission spectra at 548 nm (Tb) and 612 nm (Eu) were enhanced, as compared to the emission from the pure complex. Furthermore, the fluorescence lifetimes ranged from 0.6 to 1.5 ms and the quantum yields ranged from 32% to 61%. The luminescent fibrous membranes have potential applications in the fields of display panels, innovative electronic and optoelectronic devices.

## 1. Introduction

Rare earth (RE) ions, especially lanthanide (Ln) ions, have particularly excellent luminescence characteristics with extremely sharp emission bands, making them attractive for use in technological applications, such as optoelectronics devices [1,2,3,4,5] and sensors [6,7]. These properties arise from their 4f–4f electronic transitions. Eu(III) emits at 612 nm and has tremendous commercial importance as a red phosphor. Tb(III) emits at wavelengths between 540 and 560 nm, and could provide a viable technology for green phosphors. It is well established that in RE complexes with organic ligands, the emission from RE ions is ascribed to the effective energy transfer from the triplet state of the ligand to the crystal field states of the central RE ions [8]. Both Eu(III) and Tb(III) have 6 unpaired f-electrons, and exhibit strong luminescent emissions. The pertinent 4f–4f electronic transition of interest is ^5^D_0_→^7^F_J_ for Eu(III), with *J* = 0,1,2; whereas ^5^D_4_→^7^F_5_ is the strongest transition for Tb(III) at 546 nm. Energies of the most stable free-ion quintet excited states reveal that ^5^D_4_ [i.e., ≈20,000 cm^−1^ for Tb(III)] is significantly higher than ^5^D_0_ [i.e., ≈17,400 cm^−1^ for Eu(III)], which partially explains why Tb(III) complexes emit green photons and Eu(III) complexes emit red photons.

The direct absorption of Ln^3+^ ions is weak, so the Ln^3+^ emissive state is achieved through the excitation of a coordinated organic ligand and the subsequent energy transfer from its triplet state to the metal ions by a dipole-dipole exchange mechanism [9]. However, these complexes cannot be used directly in some practical applications due to poor thermal stability and weak mechanical properties, which limit their durability and processability. These weaknesses can be overcome by doping RE complexes into inorganic matrices. Inorganic hosts have stable physicochemical properties and provide excellent protection from environmental challenges, including photo, thermal and mechanical attacks. The encapsulation of lanthanide complexes in organic and inorganic matrices can also enhance luminescence by confinement and immobilization, which may reduce nonradiative mechanisms of excited state decay.

Electrospinning is a simple and low-cost method for the continuous production of nanofibers [10] that was made known by Formhals [11]. Electrospinning has been used to develop new lanthanide-containing luminescent materials [12]. The electrospinning method may be used to rapidly produce materials with controlled composition and organization of lanthanides, to achieve tunable properties and enhanced luminescence. For example, Hong Shao et al. reported electrospun poly(methyl methacrylate) fibers containing Eu(III) and Tb(III) complexes [13]. Their work focused on color tunability, achieved by using different ratios of Eu(III) and Tb(III) complexes in the electrospinning solution. M. K. Abd-Rahman and N. I. Razaki described an innovative multilayered membrane based on Tm(III)-containing alternating layers of thin films and nanofibers made of poly(vinyl alcohol) and Tm(III)-doped SiO_2_-Al_2_O_3_ [14]. They demonstrated enhancement of the Tm(III) luminescence, which they partially attributed to the confinement of the Tm(III) ions in the one-dimensional nanofiber environment. The properties of Tb(III) and Eu(III) have been studied in our research group. Xiaolin Zhang [15] demonstrated red light emitting electrospun nano-PVP fibers that incorporated novel three-layer Ag@SiO_2_@Eu(tta)_3_phen nanoparticles. The corresponding luminescent intensity (612 nm) of the Ag@SiO_2_@Eu(tta)_3_phen-NPs is enhanced up to 10 times compared with the pure Eu(tta)_3_phen complex. Jianhang Shi [16] investigated a novel anhydrous preparation of silica (SiO_2_)-encapsulated Tb^3+^ complex nanoparticles. The SiO_2_-Tb^3+^ nanoparticles are incorporated in electrospun polyvinylpyrrolidone hybrid nanofibers. There is an increase in the fluorescence intensity of SiO_2_-Tb^3+^ nanoparticles compared with the pure Tb^3+^ complexes. In addition, the influence of pH on the fluorescence of Tb^3+^ complexes was described.

In this paper, we propose a new fluorescent nanofiber membrane obtained by electrospinning doped with highly sensitive fluorescent complexes. This strategy combines the outstanding optical properties of RE ions with the ease of production of polymer nanofibers. The electrospinning device used in this work consists of three parts: a high-voltage supply, syringe pump filled with polymer solution and a collection electrode, as shown in Figure 1. The polymer solution is pumped through a syringe with a needle. The spinning solution forms droplets suspended from the nozzle under the effect of gravity, the solution viscosity and the surface tension. When high voltage is applied between the tip and the collection electrode, a “Taylor” [17] cone, named after Geoffrey Taylor, is achieved. The solution is drawn from the Taylor cone towards the collection electrode. As the solution travels towards the collection electrode, the solvent rapidly evaporates, precipitating the polymer into a fiber [18]. The change in the mechanical properties of the fiber from a viscous solution to an elastic solid as it travels from the tip to the collector results in a “bending instability,” which cause the fiber to whip and further stretches it [19,20,21]. Four different complexes in the fluorescent hybrid nanofibers, i.e., Tb(acac)_3_phen, Tb(acac)_3_phen@SiO_2_, Eu(tta)_3_phen, and Eu(tta)_3_phen@SiO_2_, are used as the luminescent emitters, while poly(vinylidene fluoride) (PVDF) is selected as the protective matrix for the rare earth complexes. The morphology and fluorescence properties of as-prepared nanofibers are reported. These hybrid nanofibers have potential applications in the fields of optical devices [22] and sensor systems [23].

## 2. Experimental

### 2.1. Materials

Poly(vinylidene fluoride) (PVDF, *Mw* = 534,000, Kynar 720, Arkema, Colombes, France) was used without further purification. *N,N*-Dimethylformamide (DMF, >99.5%, *Mw* = 73.09, AR), acetone, TbCl_3_ (99%), acetylacetone (acac > 99%, *Mw* = 100.12, AR), 1,10-phenanthroline (phen, *Mw* = 198.22, AR), EuCl_3_ (99%), ammonia (25–28%,*Mw* = 17.03, AR), 2-thenoyltrifluoroacetone (tta, 99%, *Mw* = 222.2), tetraethyl orthosilicate (>28.4%, *Mw* = 208.33, AR), and NaOH (*Mw* = 40, AR) were purchased from China National Medicines Group (Beijing, China).

### 2.2. Preparation of Solution of Tb(III) and Eu(III) Complexes, and Lanthanide-Doped Silica

The SiO_2_ precursor solution was prepared by mixing 20 mL ethanol, 2 mL water and 1.6 mL ammonia, stirring for 30 min, and then adding tetraethyl orthosilicate and hydrolyzing for 12 h at room temperature. To prepare the Tb(III) complexes, acetylacetone (acac, 6 mmol) and phen (2 mmol) were dissolved in 20 mL of ethanol with continuous magnetic stirring for 2 h. TbCl_3_ (2 mmol) was dissolved in 20 mL of ethanol in another beaker with magnetic stirring for 2 h, and then added to the acac and phen solution. The combined solution was continuously magnetically stirred for 4 h. The pH of the complex solution was adjusted to 8–9 by the addition of sodium hydroxide, to obtain the complex precipitate. The precipitate was separated by filtration and dried for 4 h at 40 °C in an oven to obtain a powder of Tb(acac)_3_phen complex. The synthetic procedure for Tb(acac)_3_phen along with its chemical formula is shown in Figure 2.

A similar procedure was followed for the preparation of Eu(III) complexes. Solutions of tta (0.333 g, 1.5 mmol) and phen (0.099 g, 0.5 mmol) were prepared in 5 mL of ethanol with continuous magnetic stirring for 2 h. EuCl_3_ (0.129 g, 0.5 mmol) was dissolved in 5 mL of ethanol in another beaker with magnetic stirring for 2 h. The EuCl_3_ solution was added to the tta and phen solution, and the combined solution was continuously magnetically stirred for 4 h. The pH of the complex solution was adjusted to 8–9 by the addition of sodium hydroxide to obtain the complex precipitate. The precipitate was separated by filtration and dried for 4 h at 40 °C in an oven to obtain a powder of Eu(tta)_3_phen complex. The synthetic procedure for Eu(tta)_3_phen, along with its chemical formula, is shown in Figure 3.

To prepare Tb(acac)_3_phen-doped SiO_2_ and Eu(tta)_3_phen-doped SiO_2_, the complexes were first dissolved in ethanol at the same concentrations as the original synthesis solutions (50 mmol L^−1^ in 40 mL for Tb(III) and 50 mmol L^−1^ in 15 mL for Eu(III)). Then, the complex solutions were combined with the silica solutions for 4 h with magnetic stirring. The solution was centrifuged for 10 min at a rate of 10,000 rpm to collect the complexes, and dried for 4 h at 40 °C in an oven to obtain the Si-Tb^3+^ (Silica doped into the complex of rare earth terbium) and Si-Eu^3+^ (Silica doped into the complex of rare earth europium) as dried powders.

### 2.3. Preparation of Fibrous Membranes by Electrospinning

PVDF solutions were obtained by dissolving 1 g PVDF powder in 10 mL mixed solvent of acetone and DMF in a volume ratio of 3 to 7 ((v/v) = 3:7) with magnetic stirring for 12 h. To prepare the samples containing the lanthanide complex-containing silica, the lanthanide complex-containing silica powders (0.01 g) were dissolved in 1 mL DMF and then added to the PVDF solution, so that the final PVDF solution was the same concentration, with the same 3:7 ratio of acetone to DMF. The homogenous solution was added to the 5 mL syringe which was placed in a syringe pump. The nanofibers were produced by an electrospinning apparatus (DFS-01, manufactured by Beijing Yuweixin Technology Park Co., Ltd., Beijing, China). A positive voltage of 18 kV was applied to the needle, and the solution was pumped at 2 mL/h. The distance between the tip of the needle and grounded collector was 16 cm. A pure PVDF membrane, M-Si-Tb^3+^ (the PVDF membrane with Si-Tb^3+^ powders) and M-Si-Eu^3+^ (the PVDF membrane with Si-Eu^3+^ powders) were obtained.

### 2.4. Characterization

The topographic analysis and roughness evaluations were performed in contact mode using an A.P.E. Research A100-SGS atomic force microscope (AFM, Seiko, Japan). Data acquisition and image processing were performed with the help of NanoScope Analysis (Japan, NanoScope Analysis 1.7). Scanning electron microscope (SEM) images of the membrane were obtained using a SIGMA 500/PV (SIGMA Inc., St. Louis, MO, USA), with the electron microscope operating at 200 kV. Fluorescence spectra, lifetime and quantum yield were recorded on a photo counting spectrometer from Edinburgh FLS1000 steady-state transient fluorescence spectrometer (Edinburgh Inc., Livingston, UK) with microsecond pulse lamp as the excitation. Thermogravimetric analyses (TGA, Q50, Waters LCC) were used to examine the degradation temperature of the pure PVDF powders. The analysis of PVDF membranes (Q50, Waters LCC) was carried out in a temperature range of 20–1000 °C with a scan rate of 10 °C/min under 50 mL/min N_2_ gas flow. Differential scanning calorimetry (DSC) was used to determine the melting temperature and the heat of fusion of the electrospun fiber (Q20, Waters LCC). The samples were scanned from 20 to 250 °C at a rate of 10 °C/min under 50 mL/min N_2_ gas flow.

## 3. Results and Discussions

### 3.1. Morphology of the Electrospun Fiber Polymer Membranes

The pure PVDF membrane obtained by electrospinning is shown in Figure 4. The pure PVDF membrane exhibits high transparency and flexibility, i.e., the words on the paper can be seen clearly through the membrane.

The transparency of the membrane is not due to holes going through it, as shown in the SEM image shown in Figure 4e. The thickness of the membrane was measured using the brittle fracture section method at three different spinning times, i.e., 15, 30 and 45 min. As shown in Figure 4f–h, the average thickness is 7.7, 11.1 and 16.1μm, respectively. At these thicknesses, the membrane appears purely transparent. The surface structure of the PVDF nanofibrous membrane was measured, and a quantitative surface roughness analysis was performed using AFM, as shown in Figure 5. An air-dried membrane sample was fixed on a specimen holder and 2.5 μm × 2.5 μm areas were scanned by tapping mode in air at room temperature. The roughness of samples (a), (b) and (c) are 40.9, 15.2 and 8.36 nm, making them smoother than the PVDF films doped with particles reported in other articles [24,25]. Electrospinning was conducted with the needle in either a fixed position (fixed-point) or moving at a constant horizontal rate (sweeping). The sweeping method results in reduced roughness compared to the fixed-point method. The roughness can be further reduced by drying the samples at 100 °C, which removes residual solvent.

PVDF has low surface energy, high thermal stability and hydrophobicity, which make it an attractive material for membranes [26,27,28,29,30]. Previous studies have developed hydrophobic or superhydrophobic PVDF membranes through the introduction of inorganic particles that increase the roughness and/or reduce the surface energy [24,31]. During the electrospinning process in this study, high voltage, solvent volatilization and stretching of PVDF lead to the formation of the β-phase, as shown in Figure 6, which can improve the performance of PVDF membrane materials [32,33,34]. PVDF has four phases (α, β, γ and δ), among which the β-phase shows more favorable properties, including higher melting and degradation temperatures [35]. The electrospun PVDF membrane was compared to a powder sample to determine phase transition using TGA and DSC, as shown in Figure 7. The PVDF powder starts degrading at approximately 405 °C, while the electrospun membranes do so at 415 °C. The mass loss is about 70%, and the remaining mass is carbon, which is consistent with the atomic composition. Figure 7b compares the melting endotherm areas and the crystallization exotherm areas upon cooling. The PVDF membrane melting temperature peak (at ~163 °C) is higher than PVDF powder melting temperature (at ~160 °C), and the membrane crystallization temperature (at ~130 °C) is higher than the powder crystallization temperature (at ~120 °C) due to close packing and greater orientation of the polymer chains following electrospinning. We can conclude that some crystal phases underwent a crystal transformation from the α phase to the β phase.

### 3.2. Fluorescent Properties of Ln, Ln^3+^-Si M-Ln^3+^ and M-Ln^3+^-Si

According to Crosby’s [36] model of the luminescence mechanism of RE ions and their organic complexes, the ligand absorbs a photon and is converted from the stable state (S_0_) to the excited singlet state (S_1_). The ligand subsequently crosses to the excited triplet state (S_1_-T_1_). When the triplet state is equivalent to or slightly higher than the excited state of Tb^3+^ (^5^D_4_), a nonradiative energy transfer process (T_1_-^5^D_4_) is generated, which transfers energy to the Tb^3+^ ion. Then, characteristic fluorescence is emitted when the fluorescent rare earth ion returns from the excited state to ^7^F*_J_* (*J* = 3, 4, 5, 6). According to the Dexter’s [37] solid-state sensation theory, the probability of rare earth ion radiation transition depends on the degree of matching between the triplet energy level of the ligand and the excited energy level of the rare earth ion. Sato et al. [38] put forward that when the difference (△E) between the triplet state and the energy level of the relevant Tb (^5^D_4_) is 2100–2700 cm^−1^, the organic ligand is most sensitive. For the acetylacetone complex, *T*_1_ = 25,310 cm^−1^ (acac), *S*_1_ = 20,500 cm^−1^ (^5^D_4_, Tb), and △*E* = 4810 cm^−1^. The incorporation of phen reduces vibrational relaxation due to the inclusion of three benzene rings, as shown in Figure 2. According to the Jablonski energy level diagram for free fluorescence molecules that have no other quenching processes, the lifetime τ and quantum yield $Q_{m}$ of a fluorophore are given by [39]
(1)$$\tau =1/(\Gamma +k_{nr})$$
(2)$$Q_{m}=\Gamma /(\Gamma +k_{nr})$$

It can be seen from the above formulas that higher rates of radiation decay (Γ) lead to higher quantum yields (*Q_m_*) of fluorescent molecules and shorter lifetimes (τ). For a given fluorescent molecule, the rate of radiation decay, Γ, is an intrinsic constant. The main reason for the observed spectral changes in fluorescence molecules caused by quenching or resonance energy transfer is that these processes provide more nonradiative decay pathways for excited-state fluorescent molecules. Therefore, to increase the quantum yield, increasing the fluorescence signal can only be achieved by reducing the rate of nonradiative decay.

Figure 8a shows that there are four main emission lines peaking at about 491, 547, 588 and 622 nm, corresponding to the 4f–4f transitions (^5^D_4_→^5^F_6_, ^5^D_4_→^7^F_5_, ^5^D_4_→^7^F_4_ and ^5^D_4_→^7^F_3_) of Tb^3+^ ions, respectively, which is consistent with the previous literature [8,9,40,41]. The inset graph represents the excitation schematic. The ^5^D_4_→^7^F_5_ transition is very intense at λ = 547 nm, which is responsible for the green emissions observed with these samples. The electric dipole transition ^5^D_4_→^7^F_5_ is a so-called hypersensitive transition [42]. The presence of PVDF generally increases the fluorescence intensity of the ^5^D_4_→^7^F_5_ of Tb^3+^ ions. When the Tb complexes are incorporated into the microcavities of the polymer matrix, the complexes exhibit more disordered local environments due to the influence of the surrounding polymer. Under the influence of the electric field of the surrounding ligands, the distortion of the symmetry around the lanthanide ions by the polymer results in the polarization of the ions, which increases the probability for electronic dipole transitions [43]. Therefore, the fluorescence intensity of the Si-Tb^3+^ is significantly higher than that of the undoped Tb^3+^ complexes. This may be due to the fact that the silica plays a significant role in promoting improvement of the luminescent properties of the terbium hybrid materials. After electrospinning, the fluorescence of the membrane obtained is higher than that of the complex.

To determine the lifetimes of the terbium hybrid materials, fluorescence decay curves are measured; the results are shown in Figure 8b, and the data of the fluorescence, lifetime and quantum yield are shown in Table 1. The fluorescence lifetime of the Tb^3+^, Si-Tb^3+^, M-Tb^3+^ and M-Si-Tb^3+^ are 1404.8, 1272.1, 1029.3, 999.5 μs, respectively. These fluorescence lifetimes greater than 1 ms are largely improved compared to those described in previous reports [44,45,46]. The quantum yields range from 41% to 61%. Increasing fluorescence corresponds to decreased lifetime. The fluorescence intensity and lifetime are not directly proportional. The fluorescence lifetime indicates the average time for which the particles are present in the excited state, and the fluorescence intensity is related to the photons emitted by the ligand after the absorption energy is transferred to the RE ions. The more rigid environment of the silica results in fluorescence enhancement when the complexes are doped into silica. Furthermore, the β-PVDF phase can increase the stability after electrospinning. Both effects reduce the rate of nonradiative decay, as structural rigidity reduces the vibration-related modes of energy consumption and increases the efficiency of the energy transfer. To sum up, the introduction of rigid materials can largely improve the luminescence of the hybrid materials by reducing the rate of nonradiative decay.

For the Eu(tta)_3_phen complex, *T*_1_ = 19,050 cm^−1^ (tta), *T*_1_ = 18,240 cm^−1^ (phen) and the *S*_1_ of the Eu is 16,340 cm^−1^. The energy is transferred from the tta to the phen. The complex undergoes a nonradiative transition from the *T*_1_ to an excited state of the Eu^3+^ ion due to the difference (△*E*) between the *T*_1_ state and the energy level of the relevant Eu (^5^D_4_), which is 1900 cm^−1^. The corresponding emission spectra are measured for the europium hybrid materials. The emission lines assigned to the transitions ^5^D_0_-^7^F_1_, ^5^D_0_-^7^F_2_ and ^5^D_0_-^7^F_3_ are at about 580, 612 and 650 nm, as shown in Figure 9a [47,48]. The emission intensities of the electric dipole transition ^5^D_0_-^7^F_2_ are the strongest at wavelength λ = 612 nm, pointing to a highly polarized chemical environment around the Eu^3+^ ion that is responsible for the brilliant red emission of these samples. The typical decay curves are shown in Figure 9b; the fluorescence intensity is very strong, which is consistent with the photograph in Figure 4d. The fluorescence lifetimes of the Eu^3+^, Si-Eu^3+^, M-Eu^3+^ and M-Si-Eu^3+^ samples are 754.4, 732.7, 670.4 and 634.1 μs (Table 2), respectively, corresponding to previous reports [47,49]. The quantum yield ranges from 32% to 36%. The trend of fluorescence changes for the different Eu(III) samples are consistent with those observed for Tb(III).

## 4. Conclusions

Transparent luminescent membranes containing Tb^3+^ and Eu^3+^ complex-doped silica nanoparticles were prepared via electrospinning PVDF. We investigated the effect of encapsulating these complexes in silica, and incorporating them in nanofibers, on the fluorescence. To reduce the nonradiative decay, the inorganic molecular silica was introduced into the complex. The fluorescence intensities and the quantum yields were enhanced. After electrospinning, the fluorescent intensity was enhanced further. We obtained luminescent hybrid materials with high fluorescence intensity and quantum yield. The luminescent fibrous membranes have potential applications in the fields of fluorescent clothing, counterfeiting and labels, and in the development of innovative electronic and optoelectronic devices. The membrane can absorb UV light and emit visible light; as such, our research team is conducting experiments and research on the application of this fluorescence membrane in UV filtration.

## Figures and Tables

**Figure 1 nanomaterials-10-00694-f001:**
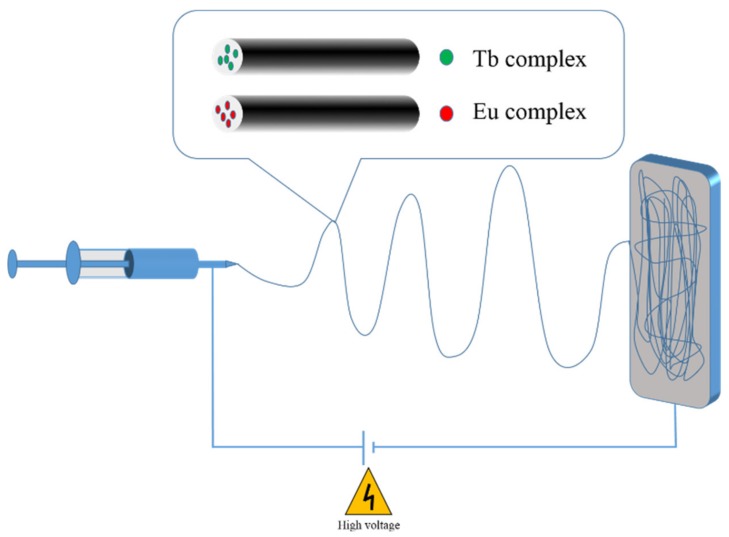
Schematic illustration of the electrospinning apparatus.

**Figure 2 nanomaterials-10-00694-f002:**
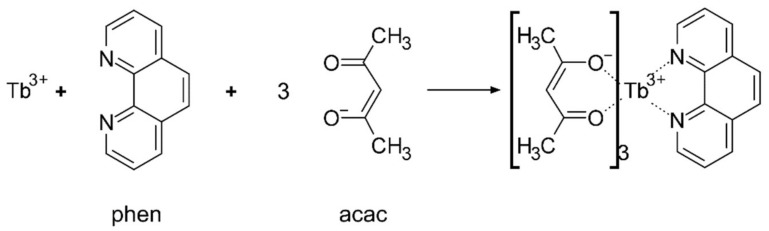
Synthetic procedure for Tb(acac)_3_phen.

**Figure 3 nanomaterials-10-00694-f003:**
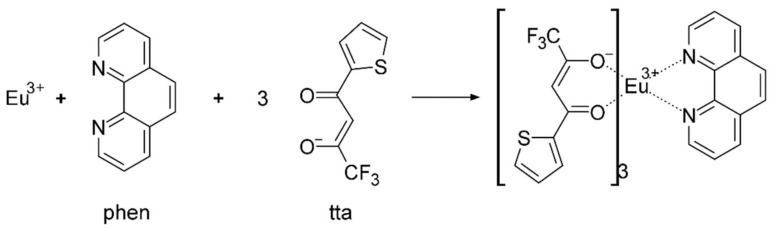
Synthetic procedure for Eu(tta)_3_phen.

**Figure 4 nanomaterials-10-00694-f004:**
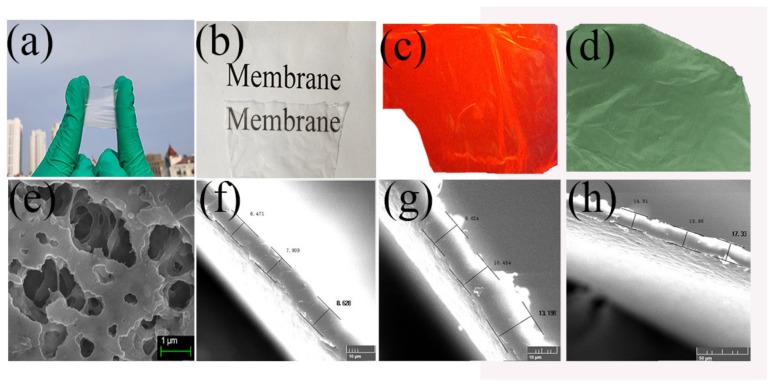
Photographs of pure PVDF complex membrane under daylight (**a**) and (**b**), and UV light (**c**) and (**d**); SEM image of PVDF electrospinning membrane (**e**) Surface structure; (**f**–**h**) Cross-section structure.

**Figure 5 nanomaterials-10-00694-f005:**
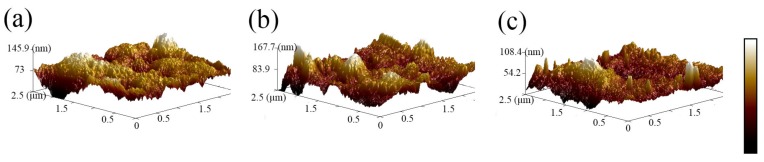
AFM of surface for pure PVDF nanofibrous membranes (**a**) fixed-point spinning (**b**) sweeping spinning and (**c**) fixed-point spinning and drying at 100 °C.

**Figure 6 nanomaterials-10-00694-f006:**
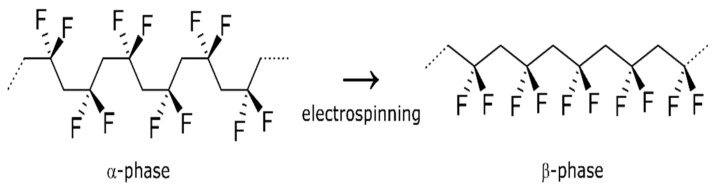
Changes in PVDF molecular structure during electrospinning.

**Figure 7 nanomaterials-10-00694-f007:**
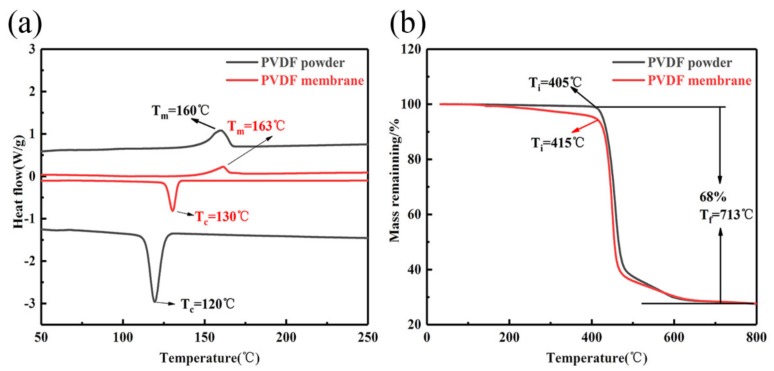
(**a**) TGA curve for pure PVDF powder and PVDF membrane after electrospinning; (**b**) DSC thermograms of pure PVDF powder and PVDF membrane after electrospinning.

**Figure 8 nanomaterials-10-00694-f008:**
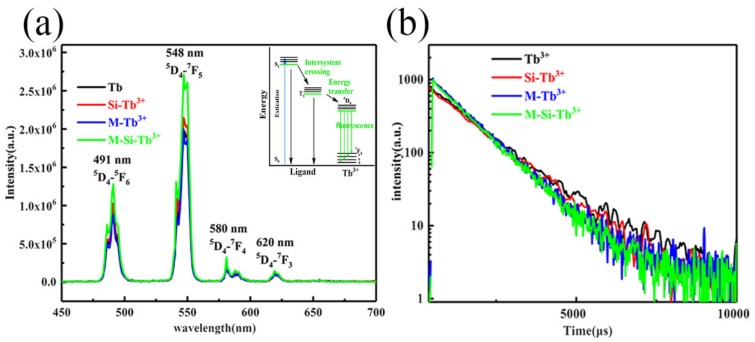
(**a**) Emission spectra (λ*_ex_* = 327 nm) and internal energy transfer mechanism of Tb^3+^, Si-Tb^3+^, M-Tb^3+^ and M-Si-Tb^3+^; (**b**) the decay curve of fluorescence lifetime.

**Figure 9 nanomaterials-10-00694-f009:**
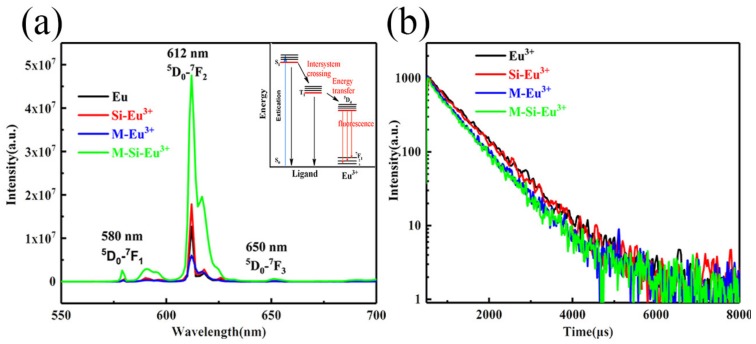
(**a**) Emission spectra (λ*_ex_* = 385 nm) and internal energy transfer mechanism of Eu^3+^, Si-Eu^3+^, M-Eu^3+^ and M-Si-Eu^3+^; (**b**) the decay curve of fluorescence lifetime.

**Table 1 nanomaterials-10-00694-t001:** Data on lifetime and quantum yield of Tb^3+^, Tb^3+^-Si M-Tb^3+^, M-Tb^3+^-Si.

Sample	Life Time (μs)	Quantum Yield (%)
Tb^3+^	1404.8	41
Si-Tb^3+^	1272	43
M-Tb^3+^	1029	46.9
M-Si-Tb^3+^	999	61.5

**Table 2 nanomaterials-10-00694-t002:** Data on lifetime and quantum yield of Eu^3+^, Si-Eu^3+^, M-Eu^3+^, M-Si-Eu^3+^.

Sample	Life Time (μs)	Quantum Yield (%)
Eu^3+^	754.4	32
Si-Eu^3+^	732.7	34
M-Eu^3+^	670.4	35
M-Si-Eu^3+^	634.1	36.8

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
