# Peer review of "Tb3+/Eu3+ Complex-Doped Rigid Nanoparticles in Transparent Nanofibrous Membranes Exhibit High Quantum Yield Fluorescence"

_nanomaterials, 2020, doi:10.3390/nano10040694_

Round 1

Reviewer 1 Report

Using PVDF electrospinning, the authors have reported the fabrication and characterisation of transparent membranes Tb3+ complex and Eu3+ complex-doped silica nanoparticles. In general, the manuscript is not well written, and the figures have poor qualities. Firstly, the introduction needs to be rewritten, and the authors need to better explain the novelty of their work. Secondly, it is unclear why most of the provided results are for pure PVDF nanofibrous membranes. The authors need to thoroughly characterise transparent membranes fabricated from the Tb3+ complex and Eu3+ complex-doped silica nanoparticles (please refer to the characterisation procedures conducted by this paper:  https://doi.org/10.1007/s10544-017-0215-y). Most importantly, the results of mechanical strength, porosity and SEM images of the fabricated electrospun membranes need to be provided.

Minor issues:

  • In the abstract, the authors included the following statement without first refereeing to the role of poly(vinylidene fluoride) (PVDF) in this study.

“During the electrospinning process, the PVDF transitions from the α phase to the β phase, which exhibits a more rigid structure.”

  • Fig 4: Please provide the actual images of the membranes fabricated from the silica
  • Table 1 and Table 2: “QY” should be replaced by quantum yield.
  • Many abbreviations are not defined in the first place.

Reviewer 2 Report

Dear Authors,

In chapter 2.3, end of the paragraph, I don't understand if you want to say that 3 types are made pure PVDF, PVDF+Si-Tb3+, PVDF+Si-Eu3+. Also, you have not define what is M-Si-Tb3+ and M-Si-Eu3+.

On chapter 3.1, commenting Figure 4, you said that the membrane exhibits high transparency and flexibility and you justify it with a picture, It will be better to put numbers and say that the transmission s x% for a thickness of y um.

In the same chapter, second paragraph, you said that PVDF is beta-phase and you refers to Figure 6 but Figure 6 gives only a change in formula and did not justify your affirmation.

On Figure 7 (graph and legend), you write "power" for "powder".

In chapter 3.2 something is wrong between the results presented in Table 1 and 2 and your comments. The way the quantum yield behave did not follow the relation (1) and (2) and your comments. Following your comment shorter lifetime correspond to higher quantum yield and correspond to relation (1) and (2) but it is not what is presented in the tables. So your explanations must revised and new explanation or new experiments must be made.

Your conclusion must be revised in function of your explanation. The part on the applications could be enriched to explain wha is the interest of this membrane in the different applications.

Best regards

Round 2

Reviewer 1 Report

The authors have improved the quality of their paper to some extent but did not conduct additional experiments due to the current pandemic situations.

 It is still confusing why there is not a proper comparison among the various fabricated membranes, i.e., pure PVDF, PVDF+Si-Tb3+, PVDF+Si-Eu3+. Specifically, why figures 4 and 5 are for “pure” PVDF membranes?

The authors need to illustrate properly (both qualitatively and quantitatively) the effect of doping Tb3+ complex and Eu3+ complex-doped silica nanoparticles into the PVDF membranes (i.e., before and after the addition of the nanoparticles).

Reviewer 2 Report

Chapter 3.1: You write,It is not a communication hole.....

I would have preferably said: "the transparency of the membrane is not due to holes going through the membrane as shown in figure 4(e) SEM image"
